# Is the Air Too Polluted for Outdoor Activities? Check by Using Your Photovoltaic System as an Air-Quality Monitoring Device

**DOI:** 10.3390/s21196342

**Published:** 2021-09-23

**Authors:** Simone Lolli

**Affiliations:** 1CNR-IMAA, Consiglio Nazionale delle Ricerche, Contrada S. Loja snc, Tito Scalo, 85050 Potenza, Italy; simone.lolli@cnr.it; 2Physics Department, Kent State University (Florence Campus), 800 E Summit St., Kent, OH 44240, USA

**Keywords:** aerosol, optical depth, sunphotometer, photovoltaic, solar panels

## Abstract

Over the past few decades, the concentrating photovoltaic systems, a source of clean and renewable energy, often fully integrated into the roof structure, have been commonly installed on private houses and public buildings. The purpose of those panels is to transform the incoming solar radiation into electricity thanks to the photovoltaic effect. The produced electric power is affected, in the first instance, by the solar panel efficiency and its technical characteristics, but it is also strictly dependent on site elevation, the meteorological conditions and on the presence of the atmospheric constituents, i.e., clouds, hydrometeors, gas molecules and sub-micron-sized particles suspended in the atmosphere that can scatter and absorb the incoming shortwave solar radiation. The Aerosol Optical Depth (AOD) is an adimensional wavelength-dependent atmospheric column variable that accounts for aerosol concentration. AOD can be used as a proxy to evaluate the concentration of surface particulate matter and atmospheric column turbidity, which in turn affects the solar panel energy production. In this manuscript, a new technique is developed to retrieve the AOD at 550 nm through an iterative process: the atmospheric optical depth, incremented in steps of 0.01, is used as input together with the direct and diffuse radiation fluxes computed by Fu–Liou–Gu Radiative Transfer Model, to forecast the produced electric energy by a photovoltaic panel through a simple model. The process will stop at that AOD value (at 550 nm), for which the forecast electric power will match the real produced electric power by the photovoltaic panel within a previously defined threshold. This proof of concept is the first step of a wider project that aims to develop a user-friendly smartphone application where photovoltaic panel owners, once downloaded it on a voluntary basis, can turn their photovoltaic system into a sunphotometer to continuously retrieve the AOD, and more importantly, to monitor the air quality and detect strong air pollution episodes that pose a threat for population health.

## 1. Introduction

Aerosols are natural or anthropogenic tiny particles, spanning several orders of magnitude (diameter ranging from microns to tens of nanometers), suspended in the atmosphere. Those short-lived climate pollutants, or near-term climate forcers, are directly emitted in the atmosphere, e.g., ashes from volcanic eruptions, black carbon from combustion processes, spray marine aerosols, desert dust, and biomass burning products, just to name few. Another category of aerosols originates from the chemical reaction of gases emitted in the atmosphere, e.g., sulphuric dioxide, nitrogen dioxide. All the different aerosol species modulate the incoming solar radiation and, to a lesser extent, the outgoing longwave Earth radiation. As an indirect effect, the aerosols interact with cloud and precipitation, especially influencing their formation and lifetime [1]. Indeed, aerosols play a crucial crucial role in cloud formation, which in turn regulate surface precipitation and then the Earth–atmosphere system radiative balance. Clouds form when the air, due to different reasons, is cooled and it is supersaturated with respect to ice or water (the only exception is represented by the so-called “funnel clouds”). The vapor in excess in the atmosphere, because of a high-energy barrier, is not capable by itself to condensate to form clouds. The presence of aerosols helps the condensation process because these tiny particles act as nuclei where water vapor can condensate over. The cloud drop size is strictly dependent on the atmospheric aerosol number density. In turn, the cloud drop size is influencing other processes such as coagulation and condensation. For this reason, the presence of aerosols in the atmosphere influences both cloud formation and dissipation and precipitation events. A recent research by Zheng et al. [2] assesses the impact of higher aerosol atmospheric concentrations on precipitation intensity and duration over the Beijing metropolitan area. The study’s main finding states that higher aerosol concentrations are linked to 25% lower precipitation frequency and 14.8% lower precipitation duration, while a higher precipitation intensity of 13.5% is found on average on higher-pollution days with respect to clear conditions.

Another important aspect that should not be underestimated is that the aerosol effects on climate are still poorly understood, as highlighted by the Fifth Assessment Report (AR5) of the Intergovernmental Panel on Climate Change (IPCC) [3]. For example, large amounts of black carbon are produced and injected into the atmosphere every year by wildfires [4], which have become more and more numerous worldwide. As a consequence, transported black carbon at polar latitudes and over mountains deposits both on ice and snow, contributing to accelerating their melting [5]. This happens because the snow and ice albedo is lowered by the presence of soot and much more solar radiation is absorbed. On other hands, the black carbon, also generated from incomplete combustion of fossil fuels and biomass burning, is considered as greatly responsible for cardiovascular and respiratory illness [6,7]. In [7], the authors found a direct relationship between high exposure to black carbon and a higher blood pressure that can lead to strokes and cardiovascular disease. As previously stated, even if the AOD is a column variable, it can still be used as a proxy to evaluate the concentration of the particulate matter with a diameter smaller than 2.5 μm (PM${}_{2.5}$) at ground [8] and then detect dangerous air-pollution episodes that can pose a serious health threat to the population [9].

Due to the multiple reasons mentioned above, the scientific community have put a lot of effort in the past decades to quantitatively assess the aerosol role, especially in the Earth–atmosphere radiative budget. For this reason, the optical, microphysical and geometrical aerosol properties are assessed and monitored mostly through remote sensing techniques, both from satellite [10], aircraft and ground-based platforms [11]. Burton et al. [12] pioneered studies of aerosol characteristic and speciation by lidar observations from aircraft. Ryder et al. [13] report the results of Fennec measurement campaign that took place in 2011 to study the microphysical properties of the Saharan dust outbreaks by aircraft measurements. Additionally, several ad hoc measurement campaigns have taken place in particular regions of the world which are extremely sensitive to climate change. Wang et al. [14] report the results of the Dongsha Island campaign, in Taiwan, that aims to characterize the properties of the transported aerosols using several months of ground-based lidar observations, while Reid et al. [15] pioneered aerosol measurements in a wild and barren region very sensitive to climate change, such as the Southeast Asia and in particular in the Philippine archipelago during the monsoon season. Similarly, Lolli et al. [16] studied how the monsoon seasons influence the aerosol atmospheric profile over Penang, Malaysia, especially in terms of radiative effects and heating rate. The study provided evidence that transboundary aerosol layers from neighboring countries are responsible for haze formation in Malaysia. Since 1998, the National Aeronautics and Space Administration (NASA) Aerosol Robot Network (AERONET [17]) has made a crucial contribution to shedding light on optical aerosol properties thanks to its global network of 2000+ sunphotometers, deployed worldwide at global scale, from equatorial to polar regions. AERONET is a proof that the atmospheric instrument networks [18] at global scale are of fundamental importance not just to study the aerosol optical, geometrical and microphysical properties, but also to validate retrievals from satellite [19,20]. The aerosol products obtained from the Moderate Resolution Imaging Spectroradiometer (MODIS) onboard the Earth Observing System’s Terra satellite launched in 1999 have been validated over more than 100 AERONET sites deployed worldwide, as reported in [21]. This first validation helped the MODIS team to improve aerosol product retrieval, especially over land. The AOD quantifies the aerosol loading present in the atmospheric column, i.e., how much light is removed from the propagation direction by scattering and absorption. As a consequence, this variable tell us how much solar radiation (wavelength-dependent) is prevented from reaching the Earth’s surface. The AOD is also a very important variable to assess the Earth–atmosphere radiation budget. More recently, it is still matter of debate if aerosols can also play a role in flagging [22] or promoting transmission of COVID-19 [23]. For all those different aspects, it is then evident why AOD observations at global scale are of paramount importance both for climate and environmental studies. On other hand, the energy production by the photovoltaic systems is strongly influenced by the atmospheric conditions and of course by the concentration of aerosols in the atmospheric column. In fact, if the direct incoming solar radiation is absorbed and scattered away from the propagation direction, the ratio between direct and diffuse light changes dramatically and so does the produced electric power. Theristis et al. [24] show that the panel efficiency is strictly linked to the distribution of the solar spectrum that in turn is related to the precipitable water (PW), air mass (AM) and AOD. Pairwise, Liu et al. [25] highlight that other than the meteorological variables, the Aerosol Index (AI) is critical to correctly forecast the next-day panel power output. The importance of AOD in forecasting the correct panel power and its linear correlation with the available solar radiation at surface were also pointed out by Gueymard et al. [26]. The effects of aerosol in reducing energy production are well documented. Quantitatively, results from Zhang et al. [27] highlight that in regions prone to severe pollution or dust outbreaks, the energy production is reduced by 0.15–0.31 kWh m${}^{-2}$ per day relative to clean air conditions, corresponding to a decrease of 4.8–9.0%. The impact on energy production shows also an important seasonal variation. They conclude that aerosol layers exert an influence similar to clouds during the winter season in northwestern China, with a drop in energy production of 11.2–17.4% in December. The authors of [28,29] also report similar results for regions close to important dust sources (deserts). Once the technical characteristics and specifications of the photovoltaic system are known, it is possible to forecast the energy production using models that often apply machine learning techniques, as shown in [30,31].

This proof-of-concept is based on a reverse engineering process where the AOD at 550 nm is retrieved from the difference between the expected and the real produced electric power by a panel through an iterative process. The AOD at 550 nm, which is a parameter needed as input by the Fu–Liou–Gu radiative transfer model to compute the radiative fluxes at the top of the atmosphere and at surface, is increased from 0 (pristine atmosphere, AOD = 0) to a value for which the radiative transfer model computed direct, indirect and reflected fluxes used to calculate the theoretical panel power will match the real observed produced electric power within a user-defined threshold (2% in this study). In view of a future perspective, this proof-of-concept can be considered phase zero of a wider project where a smartphone application will be developed. Thanks to this application, common citizens owners of photovoltaic panels, will be able, on voluntary basis, to contribute with observation to characterize the aerosol behavior at global scale, and more important, to detect severe pollution episodes that can put at risk the more vulnerable groups, i.e., elderly people, people with underlying conditions. This methodology has several drawbacks and a limited accuracy, i.e., biases introduced by sub-visible cirrus clouds, wrong aerosol typing to name a few, but still will provide zero-cost valuable observations. The manuscript is organized as follows: after this introduction, Section 2 describes the used model to forecast the produced electric energy and the relative methodology to retrieve the AOD. In Section 3 are shown some results and validation with model. In Section 4 are reported the conclusion and future perspectives.

## 2. Model Description

### 2.1. Incoming Solar Irradiance on a Tilted Plane at Ground

The first step in retrieving the AOD from the electric power produced by a photovoltaic panel consists of computing both the solar direct and diffuse irradiance over a horizontal surface. In this study, all the radiant fluxes computations are calculated using the Fu–Liou–Gu (FLG) Radiative Transfer Model, used for several other different studies [32,33,34]. To solve the multispectral radiative fluxes at surface, the FLG model needs as input the atmospheric thermodynamic state, i.e., the vertically resolved profiles of temperature, pressure, mixing ratio, ozone concentration, other than the AOD at 550 nm. The meteorological variables and ozone concentration profiles are obtained from the mid-latitude standard atmosphere USS976 [35] standard atmosphere. The solar radiant flux at surface is computed in terms of diffuse (supposed homogeneous across the sky dome), direct and total irradiance. The FLG radiative transfer model also needs as input the Solar Zenith Angle (SZA) and the surface albedo. In this study, the photovoltaic system is located over a brick roof (see Figure 1) (right) that, according to [36], has an albedo of 0.40. For panels installed over other surfaces, i.e., downtown, or on a ship, the albedo should be changed accordingly. The second step in simulating the electric energy produced by the photovoltaic panel, consists in assessing how much of the incoming solar radiation on the horizontal plane falls into a tilted surface (representing the photovoltaic system). The radiation components that should be taken into consideration are: (i) the direct component ($I_{d}$), (ii) the diffuse component ($I_{diff}$) and (iii) the albedo-dependent reflected component by the surrounding environment ($I_{alb}$). The angle of incidence $\theta_{t}$ is defined as the angle formed between the normal to the tilted surface and the direction of the solar radiation, according to Equation (Equation 1):(1)$$\cos\theta_{t}=\cos\Phi \cos\theta_{z}+\sin\Phi \sin\theta_{z}\cos\left(\gamma -\beta \right),$$
where Φ is the panel tilting angle (0 degrees means that the panel is horizontal), $\theta_{z}$ is the solar zenith angle (SZA), γ is the solar azimuth angle, β the panel module azimuth orientation (South = 0${}^{\circ }$ degrees, positive values towards the East direction). Then the direct incoming solar radiation on a titled plane $I_{d}^{t}$ can be calculated from Equation 1 as:(2)$$I_{d}^{t}=I_{d}\frac{\cos\theta_{t}}{\cos\theta_{z}}.$$

Here, $I_{d}$ and $I_{diff}$ are computed using the previously described FLG radiative transfer model for an horizontal surface. The diffuse component on a tilted plane $I_{diff}^{t}$ is calculated considering the sky dome view factor as:(3)$$F_{sky}=\frac{1+\cos\Phi }{2},$$

$F_{sky}$ is then equal to 1 (diffuse energy from the whole hemisphere) if the panel is horizontal and 0.5 if tilted by 90 degrees (diffuse energy from half hemisphere). Then,
(4)$$I_{diff}^{t}=I_{diff}F_{sky}.$$

In a similar way, the albedo-dependent reflected component on the tilted panel is dependent on the view factor of the surrounding environment that in turn depends on the albedo. The reflected energy is dual with respect to the diffuse energy, i.e., a horizontal panel will be unaffected by reflected radiation ($F_{c}$ = 0), as shown by Equation (Equation 5):(5)$$F_{c}=\frac{1-\cos\Phi }{2}.$$

The albedo-dependant reflected component on a tilted plane is then:(6)$$I_{alb}^{t}=I_{d}\rho F_{c},$$
where ρ is the albedo, an adimensional parameter that is defined as the ratio (value 0 and 1) between the reflected solar radiation by the surrounding environment and the incoming solar radiation. As an example, fresh snow has an albedo of 0.75, while an urban environment has an albedo of 0.12, and water bodies—usually around 0.07. The total solar irradiance (in W/m${}^{2}$) falling on the tilted panel is then:(7)$$I^{t}=I_{d}^{t}+I_{diff}^{t}+I_{alb}^{t}.$$

### 2.2. Simplified Energy Production Model from a Photovoltaic Panel

Besides the incoming solar irradiance, the electric energy production is also dependent on other factors, e.g., the meteorological conditions, the photovoltaic panel efficiency. The proposed model is non-linear and takes into account the working temperature of the panels and their efficiency, retrieved from ambient wind speed and temperature.

The expected photovoltaic produced power can be expressed as:(8)$$P_{out}=P_{stc}\frac{I^{t}}{G_{stc}}\eta_{rel}\left(G^{\prime },T_{m}^{\prime }\right),$$
where $I^{t}$ is computed by the FLG model (Equation (Equation 7)). The terms in Equation (Equation 7) are found in Equations (Equation 2), (Equation 4) and (Equation 6). $P_{stc}$ represents the nominal output power at standard test conditions (1000 W m${}^{-2}$ solar irradiance and module temperature of 25 ${}^{\circ }$C). $G_{stc}$ is the reference irradiance of 1000 W/m${}^{2}$ at standard test conditions previously described. The relative efficiency is given by the empirical non-linear model:(9)$$\eta_{rel}\left(G^{\prime },T_{m}^{\prime }\right)=1+k_{1}\ln\left(G^{\prime }\right)+k_{2}\ln{\left(G^{\prime }\right)}^{2}+k_{3}T_{m}^{\prime }+k_{4}T_{m}^{\prime }\ln\left(G^{\prime }\right)+k_{5}T_{m}^{\prime }\ln{\left(G^{\prime }\right)}^{2}+k_{6}T_{m}^{\prime }{}^{2},$$
where $k_{1},k_{2}\ldots k_{6}$ were empirically retrieved and are available in the frame of the Photovoltaic Geographical Information System project of European Commission Joint Research Center (JRC) at Ispra, Italy. For crystalline silica photovoltaic panels (as those used in this study), the k-values are reported in Table 1:

Here $G^{\prime }$ is a non-dimensional value, ratio of the irradiance $I^{t}$ and $G_{stc}$, while $T_{m}^{\prime }=T_{mod}-T_{modSTC}$ is the difference between the module operating temperature and the module operating temperature at Standard Test Conditions. A good approximation of the module temperature is given in the Equation (Equation 10):(10)$$T_{mod}=0.943T_{a}+0.028*I^{t}-1.528*V+4.3,$$
where $I^{t}$ is given by Equation (Equation 7), while $T_{a}$ is the ambient temperature and *V* is the horizontal wind speed.

### 2.3. AOD at 550 nm Retrieval

The AOD at 550 nm is retrieved through an iterative process. The AOD at 550 nm is needed as input by the Fu–Liou–Gu radiative transfer model. The iteration starts with an AOD value of 0.01. The AOD is then increased at each step by 0.01 and the direct, diffused [37] and reflected solar irradiances are the computed. The values are then injected into Equation (Equation 8), together with the photovoltaic panel technical specifications and meteorological variables to calculate the expected output power $P_{out}$. The computed value is compared with the real measured power $P_{out}^{real}$, and if the absolute difference between the two values is within a defined threshold (set at 2%), the process stops and the AOD is recorded. The algorithm, together with the FLG code, is very efficient and will run under a powerful, dedicated server that can compute hundreds of runs in seconds. The smartphone app is then also connected very efficiently to the server through an API. The processing speed is very high. In a previous study by Campbell et al. [38], more than 15,000 iterations were computed in seconds. During the retrieval process, the multiple scattering is assumed negligible, and each atmospheric layer is assumed uniform in the vertical and horizontal directions. The iterative process is detailed in the flow chart in Figure 1.

## 3. Photovoltaic Panel Output Energy Production Validation

The polycrystalline silicon photovoltaic module used in this study was manufactured by the German company Schüho (production currently discontinued) and it was installed on the roof of a private home in Tuscany, Italy (43.847 N, 10.745 E; 20 m a.s.l.). The installation site is visible in Figure 2 (Google Earth image). The photovoltaic panel technical characteristics are available at this site.

From a climatological point of view, according to the historical data from the meteorological station of Montecatini (6 km NW from the installation site), in winter, the average daily sunshine is 3.2 h and that increases to 5.2 h in spring and 8.2 h in summer. Autumn experiences 4.9 h of sunshine per day. Precipitation is mainly concentrated in autumn and winter (600 mm, 55 days of rain), while a sharp drop is witnessed during summer (142 mm, 13 days of rain). During spring, the expected rain is 254 mm (27 days of rain). The sky is overcast (8/8 oktas) for about 55% of the time in winter. This value drops to 50% in spring and to 25–30% in summer. In autumn, overcast sky rises again to 45–55%.

The hourly averaged photovoltaic panel power has been continuously measured and recorded since 2021. To calibrate the photovoltaic panel power model (Section 2.2), the AOD measurements are needed. AOD has been retrieved at different temporal and spatial scales both from space, aircraft and ground-based platforms for at least two decades. Unfortunately, AOD in situ measurements are not available at location site. A valid and effective alternative to overcome this problem is given by the Copernicus Atmosphere Monitoring Service (CAMS) of the European Centre for Medium-Range Weather Forecasts (ECMWF), which provides the total AOD values at 550 nm on a 3 h basis with a spatial resolution of 0.125 × 0.125 degrees (about 10 km × 10 km at mid-latitudes). The ECMWF-CAMS, near-real-time reanalysis, without entering in details, is based on making a retrospective AOD analysis (on a daily basis) through data assimilation from the previous day. The meteorological data needed by the model are available in real time from IMONTE142 weather station, located about 6 km NW from the residence where the photovoltaic panels are installed. The meteorological data are freely available from WeatherUnderground website. The calibration took place during 3 clear-sky days, 1 in winter and 2 in summer (Figure 3). The hourly observed and simulated (using 3 h AOD values from ECMWF-CAMS) produced powers are intercompared. Solar zenith angles greater than 70${}^{\circ }$ are disregarded from the intercomparison to avoid unwanted reflections and sun-shading from neighbour buildings and trees. The intercomparison between the observed and the forecast photovoltaic electric power shows a very good agreement, with a bias of 0.14 W/m${}^{2}$ and a Root Mean Square Error (RMSE) of 4.34 W/m${}^{2}$. The results are surprisingly good, considering all the possible different sources of error.

## 4. AOD Retrieval and Intercomparison with ECMWF-CAMS Reanalysis

### 4.1. Analysis of the Sensitivity of the Methodology

We assess the accuracy of the proposed methodology before proceeding with AOD validation and retrieval. A sensitivity study is performed on the FLG model by varying the input parameters. The major sources of uncertainty might be summarized as follows: (i) deterioration of photovoltaic module with respect to the expected technical characteristics, (ii) the use of standard atmospheric models, e.g., USS976, as input for the Fu–Liou–Gu Radiative Transfer Model instead of the current meteorological variables (e.g., obtained from a radiosounding), (iii) advection of the aerosol species different from those established during the setup, (iv) presence of sub-visible cirrus clouds introducing a bias, (v) wrong value of the surface albedo where the photovoltaic panel is deployed. Other studies [24,25,26] provide evidence for how the different variables affect the performances of photovoltaic modeling, especially Air Mass (AM) and Precipitable Water (PW). The AM is exactly determined by the solar zenith angle and taken into account in the computation. The site elevation is also determined to compute the solar radiation correctly. Additionally, the model sensitivity to PW (through the inputted atmospheric water vapor mixing ratio; Figure 4) is assessed, and results are shown in Figure 5.

The FLG model sensitivity is assessed against an ideal case where the vertical aerosol load is set to AOD = 0.2 (urban aerosols) paired with summer mid-latitude USS976 atmospheric profile. Errors due to (ii) are assessed by computing the AOD with an extremely dry profile, e.g., the summer USS976 arctic profile. As shown in Figure 4, the default profile contains much more precipitable water. The analysis reveals that the cause (ii) introduces a large positive bias in the computation, above 20%. This considerable uncertainty in AOD retrieval is caused by the large discrepancies between the two water vapor mixing ratio profiles, as shown in Figure 4. This difficulty will be alleviated by incorporating the vertical profile of the water vapor mixing ratio from the global meteorological models. With respect to point (iv), the differences in diurnal AOD computation are assessed assuming the presence of a cirrus cloud with an optical depth of 0.05, a geometrical thickness of 500 m, and the cloud base at 10 km. The analysis indicates that cirrus cloud can negatively bias the AOD computations up to 13%. The difference in AOD due to a dust outbreak is assessed considering a dust layer at 4 km with an optical depth of 0.1 (half of the total AOD). In this case, a lower positive bias of 5% is found. Wrong values of surface albedo, e.g., point (v), influence the reflected solar radiation that, during clear sky conditions, is almost an order of magnitude smaller. In the new release, a look-up table will be implemented to input the correct albedo value, also using satellite observations. The characteristics deterioration of the photovoltaic panel will principally affect the computation of $P_{out}$ (Equation (Equation 8)) and depends on the photovoltaic panel model. In the future release, the final user will be able to choose the photovoltaic panel type from a menu in the app.

### 4.2. AOD Retrieval Validation

In this section I present the AOD retrievals during clear-sky measurements. The data are considered valid for the intercomparison when cloud cover is less or equal two okta. As pointed out in Chew et al. [39], invisible, tiny cirrus clouds might still be present and bias the retrieval. Please refer to Section 4.1 for an assessment of the introduced bias by those thin ice clouds. The intercomparison is presented as a scatter plot in Figure 6. Data were taken on fifteen different days from January to June 2021, from around 10 am to 1 pm local time. Data represent different conditions, from clear-sky to moderately polluted. On polluted days, the aerosols, following the atmospheric circulation, are advected towards the observational site from from large-scale industrial plants in the vicinity. It is not possible to exclude dust outbreaks affecting data (especially for larger AOD values). Dust outbreaks will bias the retrieval (see Section 4.1) as, in view of automatic retrieval from app, the user should set up the type of environment, i.e., the local background aerosol species during the installation; for this study, the setup type is urban-continental. The first prototype of a Graphical User Interface has been developed for Apple iPhone models shown in Figure 7.

## 5. Discussion

For the first time, this manuscript describes a methodology to retrieve the AOD using a photovoltaic solar system as a sunphotometer. The methodology is based on a iterative process where the AOD at 550 nm is changed as long as the theoretical produced electric power is not equal to the real produced power (within a certain threshold). This study should be considered as a “proof of concept” to demonstrate the feasibility of the proposed methodology, in view of a future development of a smartphone application that will allow photovoltaic panel owners to transform their photovoltaic module into an air-quality measurement device. Errors and uncertainties of the methodology could be high, and further tests and validation over greater temporal periods for different conditions and at different latitudes are needed. Nevertheless, despite the uncertainties, the AOD retrievals are still valuable (and most important, cost-free), both for climate studies and as a warning for bad air-quality conditions. To reduce the error sources listed in previous Section 4.1, we obtain the vertical profile of the meteorological variables at the desired location from global meteorological models such as the European Medium-Range Weather Forecast (ECMWF).

## 6. Conclusions

The presented methodology, based on a reverse-engineering process, retrieves the AOD from the hourly produced electric power by commercially available photovoltaic panels. The retrieved AOD values, intercompared with the ECMWF-CAMS reanalysis data, show a linear correlation with an RMSE of 0.033 ($R^{2}=0.94$). The presence of thin cirrus clouds, incorrect meteorological variables including vertical profiles of PW, and aerosol miss-typing can lead to a sizable bias in the retrieval up to 25%. Despite this possible large uncertainty, the presented proof of concept is part of a larger project which will allow common citizens to use their photovoltaic panels to estimate the AOD in real time. During the second phase, a smartphone application will be developed to collect the observations from all the photovoltaic panel owners to produce a global-scale AOD map. Not to be underestimated is also the smartphone application’s potential in notifying users when the atmospheric aerosol concentration is too high becoming dangerous for sensitive groups. During the setup process the user will be asked to introduce the photovoltaic panel technical characteristics, i.e., the panel tilting, its orientation with respect to the South direction the efficiency of the panels, the albedo of the surface on which the panels are installed to name few. Giving access to the position, the app will automatically retrieve the solar zenith angle, the temperature and relative humidity. There will also be the possibility to introduce those parameters manually. After the setup process, the user is required to input the current produced energy in clear-sky conditions. The app will then show and record the AOD values. Figure 7 (right) shows a notification sent to the user alerting for high pollution condition and suggesting to limit the time spent outdoors. In a future release, it will be possible to personalize the profile, e.g., elder people, sensible groups, to obtain more personalized notifications.

The methodology is based on the development of a simple model to forecast the photovoltaic panel electric power, once the technical characteristics are known. Then the clear-sky AOD is retrieved at 550 nm through an iterative process. As shown in Figure 1, the atmospheric optical depth is incremented in steps of 0.01 until the forecast produced electric energy by the model using the direct and diffuse radiation fluxes computed by Fu–Liou–Gu Radiative Transfer Model will match the real observed power by the photovoltaic system.

All the possible methodology error causes are listed in Section 5. The methodology shows large uncertainties, mostly related to the precision of the technical characteristics of the panel that might degrade with time. Moreover, meteorological parameters should be known with sufficient precision, while the Fu–Liou–Gu model needs as input the atmospheric concentration of the mixing ratio and ozone that are taken from the USS976 atmospheric model.

## Figures and Tables

**Figure 1 sensors-21-06342-f001:**
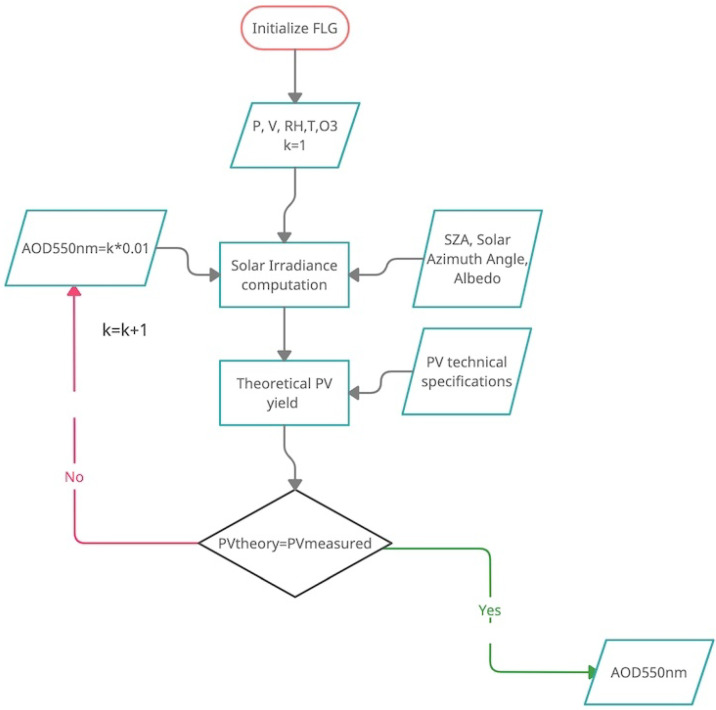
Flowchart representing the iterative process to compute the AOD retrieval at 550 nm. The FLG model uses pre-defined functions to calculate the AOD values at the different wavelengths needed for solar irradiance calculation at surface.

**Figure 2 sensors-21-06342-f002:**
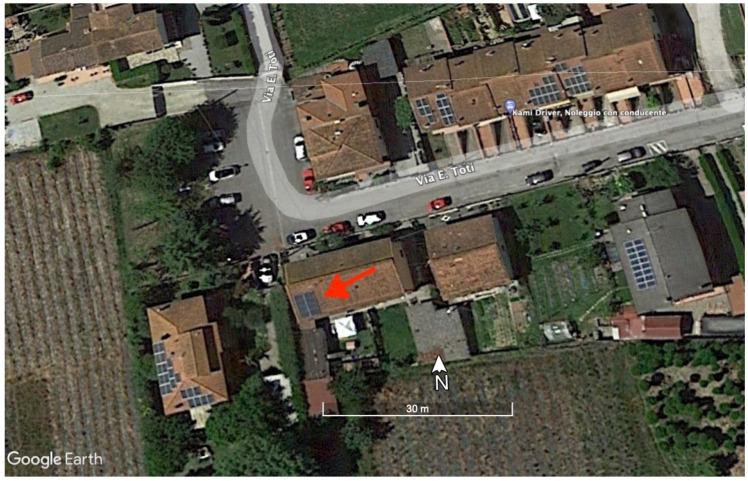
The red arrow indicates the photovoltaic modules (43.51${}^{\circ }$ N, 10.48${}^{\circ }$ E, 20 m a.s.l.) used in this study. The panel azimuth angle is 163${}^{\circ }$ azimuth and it is tilted by 30${}^{\circ }$ degrees. Image: Google Earth Pro (7.3.3.7786, 4 June 2021).

**Figure 3 sensors-21-06342-f003:**
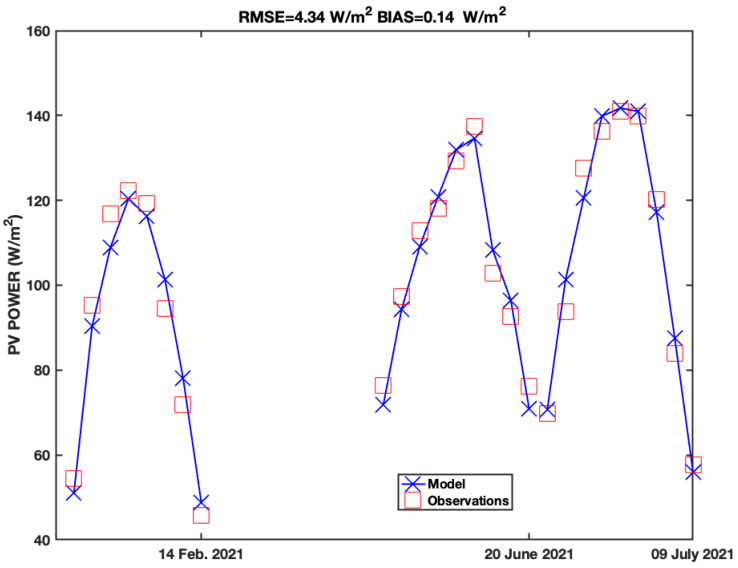
Intercomparison between the measured photovoltaic power and the power forecast by the simplified model developed in Section 2. The results show a RMSE of 4.34 W/m${}^{2}$ and a bias of 0.14 W/m${}^{2}$.

**Figure 4 sensors-21-06342-f004:**
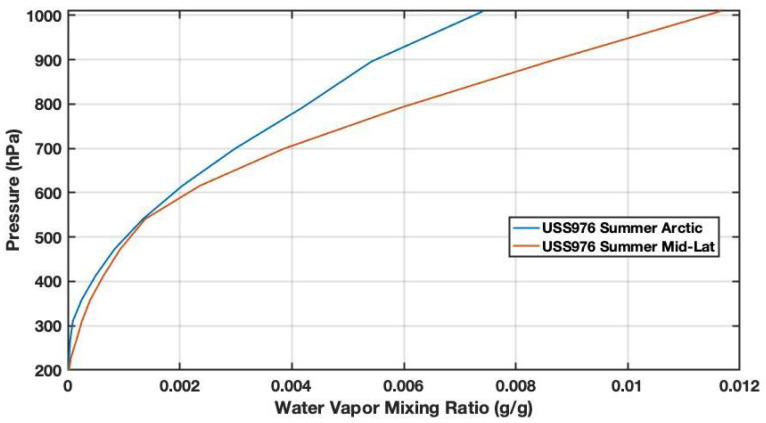
Difference between the summer mid-latitude and arctic USS976 water vapor mixing profile used to assess FLG sensitivity in computing the AOD.

**Figure 5 sensors-21-06342-f005:**
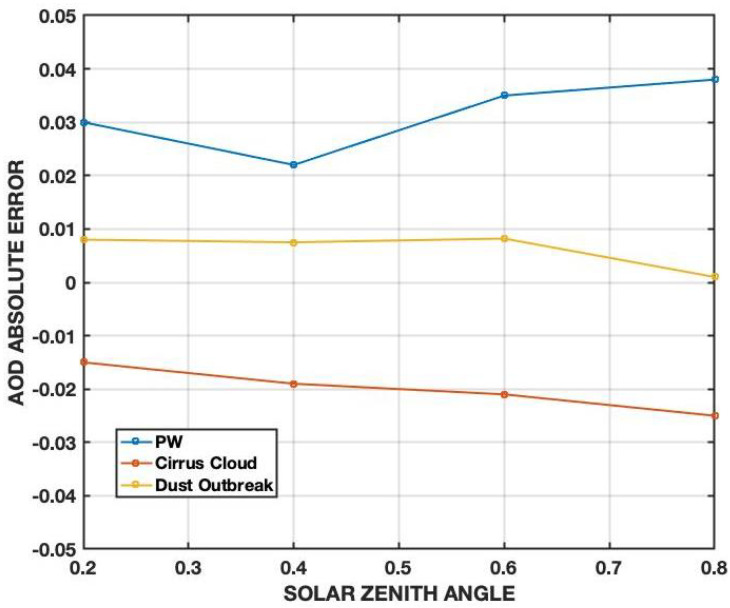
AOD absolute error vs. the Solar Zenith Angle for source of errors listed in (ii), (iii) and (iv).

**Figure 6 sensors-21-06342-f006:**
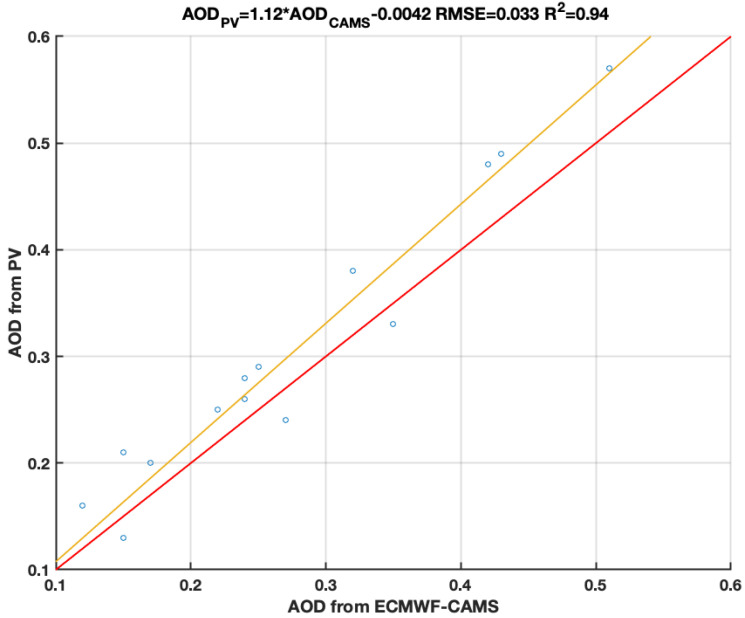
Intercomparison between AOD from ECMWF-CAMS reanalysis and AOD retrieved from photovoltaic panel power, as shown in Section 2.2. The linear fitting highlights that 94% of the variability can be explained by the model with a RMSE of 0.033.

**Figure 7 sensors-21-06342-f007:**
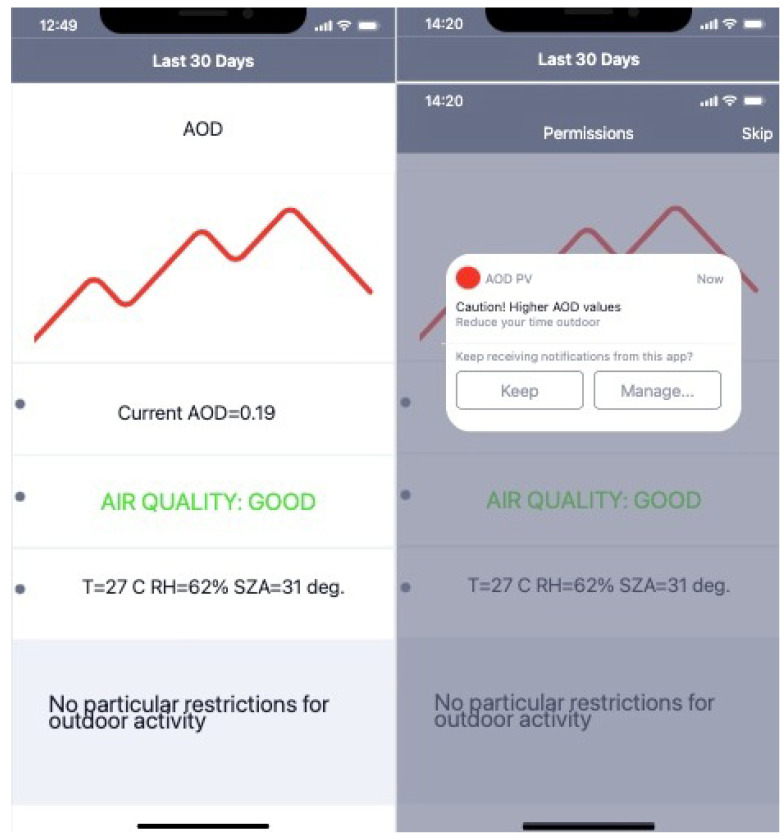
**Left**: A possible Graphic User Interface for iPhone users. The app will show the plot relative to the last 30 days of measurements. The lower panel shows the current AOD value and good air quality. **Right**: Example of notification alerting app users about pollution above a certain threshold. The notification also suggests to limit time spent outdoors. The notifications, based on AOD levels, will notify users with different kinds of suggestions.

**Table 1 sensors-21-06342-t001:** Empirical values to be used in Equation (Equation 9) for C-Si photovoltaic panels.

$k_{1}$	−0.001716
$k_{2}$	−0.040289
$k_{3}$ (${}^{\circ }$C${}^{-1}$)	−0.004581
$k_{4}$ (${}^{\circ }$C${}^{-1}$)	0.000148
$k_{5}$ (${}^{\circ }$C${}^{-1}$)	0.000169
$k_{6}$ (${}^{\circ }$C${}^{-1}$)	0.000005

## Data Availability

Meteorological data are publicly available at Weatherunderground. AOD from CAMS reanalysis are available at ECMWF. Photovolataic panel data are available upon request.

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
