# Peer review of "Is the Air Too Polluted for Outdoor Activities? Check by Using Your Photovoltaic System as an Air-Quality Monitoring Device"

_sensors, 2021, doi:10.3390/s21196342_

Round 1
Reviewer 1 Report
- The close relation between AOD and PV power has been discussed in the literature: e.g.,
- C. A. Gueymard, Daily spectral effects on concentrating PV solar cells as affected by realistic aerosol optical depth and other atmospheric conditions. In Optical Modeling and Measurements for Solar Energy Systems III (Vol. 7410, p. 741007). International Society for Optics and Photonics (2009).
- J. Liu, W. Fang, X. Zhang, and C. Yang, An improved photovoltaic power forecasting model with the assistance of aerosol index data. IEEE Transactions on Sustainable Energy, 6(2), 434-442 (2015).
- M. Theristis, E. F. Fernández, F. Almonacid, and P. Pérez-Higueras, Spectral corrections based on air mass, aerosol optical depth, and precipitable water for CPV performance modeling. IEEE Journal of Photovoltaics, 6(6), 1598-1604 (2016).
Please cite these and similar publications to summarize the current understanding of how AOD and aerosol type affects the power generated by different PV systems.
- As pointed out in the literature mentioned above, essentially, several parameters such as AOD and precipitable water (PW) can affect the PV power, in addition to the air mass and surface elevation of the site. Concerning the retrieval of AOD, the proposed method can work when the conditions are favorable, as indicated in Figs. 4 and 5. In reality, however, the reliability of this PV-based “AOD sensor” should be investigated more quantitatively by evaluating its accuracy and limitation. The sensitivity analysis should include the quantitative evaluation of the effects of the type of PV, the influence of PW, and thin cirrus clouds. Also, what is the expected accuracy of the radiative transfer code in calculating the direct and diffuse radiation components? Also, please discuss the validity of assuming a single aerosol type (and hence, the simple wavelength dependence of AOD). How about the potential outbreak of dust events, as encountered in many parts of the world?
(minor)
- The style and English usage should be brushed up. The problems include:
- The rule is that once defined, the acronym (e.g., AOD) should be used throughout the text. There are many exceptions to this rule in the current text.
- L27: “sulphuric dioxide, nitrogen dioxide” usually refer to gases, not particles.
- “Burton et al., [12]” should be “Burton et al. [12]”. Many similar cases are seen.
- Expressions such as “as reported in [21]” and “Also [25,26] report” are not recommended ways for citing references.
- The units (e.g., kWh m-2) should be roman (non-italic), and in SI, the form m-2 is preferred than /m2.
- Each equation should be part of a sentence, and hence, a comma or period should usually appear just after each equation. No indent should appear before “where” at L135.
- Please insert a space between a number and unit: e.g., “550 nm”, “10 km”, etc.
- Figs. 2 and 3 should be combined (or just delete Fig. 3). Describing the information about the PV type and weather conditions of the observation site (e.g., cloud coverage, relative humidity, seasonal rainfall rate) would be useful.
- Figure 5 shows around 15 data points. Please explain the conditions under which these points were observed (e.g., time periods, cloud coverage, etc.).
- Figs. 6 and 7 can be combined, and the figure should appear before the conclusion. Citing new figures in the conclusion section should be avoided.
- Some of the sentences are too colloquial. Please use a more formal style appropriate for a journal paper.
- The roles of the introduction section and the conclusion section must be different. Please revise the conclusion section so that it summarizes the important findings from the present work.
Author Response
I would like to thank the anonymous reviewer for the meaningful comments that greatly improve the manuscript quality. Author point-by-point answers are in red
The close relation between AOD and PV power has been discussed in the literature: e.g.,
- C. A. Gueymard, Daily spectral effects on concentrating PV solar cells as affected by realistic aerosol optical depth and other atmospheric conditions. In Optical Modeling and Measurements for Solar Energy Systems III (Vol. 7410, p. 741007). International Society for Optics and Photonics (2009).
- J. Liu, W. Fang, X. Zhang, and C. Yang, An improved photovoltaic power forecasting model with the assistance of aerosol index data. IEEE Transactions on Sustainable Energy, 6(2), 434-442 (2015).
- M. Theristis, E. F. Fernández, F. Almonacid, and P. Pérez-Higueras, Spectral corrections based on air mass, aerosol optical depth, and precipitable water for CPV performance modeling. IEEE Journal of Photovoltaics, 6(6), 1598-1604 (2016).
Please cite these and similar publications to summarize the current understanding of how AOD and aerosol type affects the power generated by different PV systems.
Thanks a lot for the suggestion. The proposed references are now integrated into the manuscript and the discussion has been expanded.
- As pointed out in the literature mentioned above, essentially, several parameters such as AOD and precipitable water (PW) can affect the PV power, in addition to the air mass and surface elevation of the site. Concerning the retrieval of AOD, the proposed method can work when the conditions are favorable, as indicated in Figs. 4 and 5. In reality, however, the reliability of this PV-based “AOD sensor” should be investigated more quantitatively by evaluating its accuracy and limitation. The sensitivity analysis should include the quantitative evaluation of the effects of the type of PV, the influence of PW, and thin cirrus clouds. Also, what is the expected accuracy of the radiative transfer code in calculating the direct and diffuse radiation components? Also, please discuss the validity of assuming a single aerosol type (and hence, the simple wavelength dependence of AOD). How about the potential outbreak of dust events, as encountered in many parts of the world?
I really appreciate the reviewer comment. I agree that this important information is missing in the manuscript. A new paragraph has been added summarizing the main finding of the analysis, putting in evidence how the AOD retrieval changes when changing the input parameters. New plots showing the AOD absolute error when changing the input variables are added in a new section. Nevertheless the implemented radiative transfer model already takes into consideration both the airmass (through the solar zenith angle) and the site elevation.
The FLG diffuse radiative component is exhaustively explained in Zhang et al., 2013: https://journals.ametsoc.org/view/journals/atsc/70/3/jas-d-12-0122.1.xml
The information has been added into the text.
(minor)
- The style and English usage should be brushed up. The problems include:
The manuscript has been revised and polished by an English native speaker.
The rule is that once defined, the acronym (e.g., AOD) should be used throughout the text. There are many exceptions to this rule in the current text.
The acronyms were defined accordingly
L27: “sulphuric dioxide, nitrogen dioxide” usually refer to gases, not particles.
The section has been rephrased and clarified.
“Burton et al., [12]” should be “Burton et al. [12]”. Many similar cases are seen.
Changed accordingly
Expressions such as “as reported in [21]” and “Also [25,26] report” are not recommended ways for citing references.
Changed accordingly
The units (e.g., kWh m-2) should be roman (non-italic), and in SI, the form m-2 is preferred than /m2.
Changed accordingly
Each equation should be part of a sentence, and hence, a comma or period should usually appear just after each equation. No indent should appear before “where” at L135.
Changed accordingly
Please insert a space between a number and unit: e.g., “550 nm”, “10 km”, etc
Changed accordingly
Figs. 2 and 3 should be combined (or just delete Fig. 3).
The figures are now combined together
Describing the information about the PV type and weather conditions of the observation site (e.g., cloud coverage, relative humidity, seasonal rainfall rate) would be useful.
The information has been added in the manuscript.
Figure 5 shows around 15 data points. Please explain the conditions under which these points were observed (e.g., time periods, cloud coverage, etc.).
The information has been added in the manuscript.
Figs. 6 and 7 can be combined, and the figure should appear before the conclusion. Citing new figures in the conclusion section should be avoided.
Changed accordingly
Some of the sentences are too colloquial. Please use a more formal style appropriate for a journal paper.
Changed accordingly
The roles of the introduction section and the conclusion section must be different. Please revise the conclusion section so that it summarizes the important findings from the present work.
Changed accordingly
Reviewer 2 Report
This is a very meaningful work. The author obtains the optical depth at 550nm from the PV system to evaluate the outdoor environmental quality, which has great application scenarios and application value.
However, there are still some issues that needed to be confirmed:
1) The initial AOD starts from 0, and the iteration step is 0.01. How efficient is this calculation? For severely polluted weather, it may take hundreds of steps to calculate, how much time it will take?
2) What is the corresponding AOD error when the iteration threshold is set to 2%? Can the uniqueness of the iteration results be guaranteed?
3) The meteorological parameters in the FLG model are obtained from the USS876 model. What is the accuracy? Can it be obtained from the local weather forecast?
4) The ground surface albedo is set to a fixed value of 0.40. Does the author consider the albedo changes in different seasons?
5) How is the predicted power obtained in Fig4? From section 2, we can see that the power is affected by multiple variables, such as temperature, pressure, AOD, etc. How to consider the change of these variables when forecasting?
6) In line 226, there is an extra “the”.
7) The iPhone app requires the user to input the power of the PV system. Does this step greatly limit the application of the method? There is no one has been staring at the output power of the PV system. Maybe the author considers integrating the calculations into the PV system and sends forecast and warning information to users’ mobile phones via the Internet. This is just a personal idea which may be a reference for the author.
Author Response
I would like to thank the anonymous reviewer for the meaningful comments that improved the manuscript. Point-by-point answers are in red
However, there are still some issues that needed to be confirmed:
1) The initial AOD starts from 0, and the iteration step is 0.01. How efficient is this calculation? For severely polluted weather, it may take hundreds of steps to calculate, how much time it will take?
Thanks for pointing this out. Of course, the more polluted atmosphere, the more computational time is needed and values up to 3 can be reached (even if the those values are uncommon). Nevertheless the FLG code is very efficient and run under a dedicated powerful server that can compute hundred of runs in seconds. The smartphone app is then connecting also very efficiently to the server through an API. As an example, the FLG code on server took about 150 seconds to analyze more than 22000 observations as shown in https://journals.ametsoc.org/view/journals/apme/60/1/jamc-d-20-0077.1.xml
2) What is the corresponding AOD error when the iteration threshold is set to 2%? Can the uniqueness of the iteration results be guaranteed?
Thanks to highlight this point. This analysis was missing in the manuscript. A new section (4.1) has been added to the manuscript that shows the results from a sensitivity study of the methodology.
3) The meteorological parameters in the FLG model are obtained from the USS876 model. What is the accuracy? Can it be obtained from the local weather forecast?
The reviewer is right using USS976 model poses a great limitation in accuracy. For a future implementation for app users, the thermodynamic variables will be obtained from global meteorological models, e.g. ECMWF, that can provide atmospheric profiles with a temporal resolution of 3 hours and a spatial resolution of 0.1 x 0.1 degrees. This information has been added in the text for future developments
4) The ground surface albedo is set to a fixed value of 0.40. Does the author consider the albedo changes in different seasons?
The reviewer is right. Albedo is changing with respect to the seasons. Please see new Section 4.1 for analysis. For the future app version, it will be implemented a look-up-table that takes into consideration changes of the albedo with time. The information has been added in the text.
5) How is the predicted power obtained in Fig4? From section 2, we can see that the power is affected by multiple variables, such as temperature, pressure, AOD, etc. How to consider the change of these variables when forecasting?
The predicted power is computed using the model in Equation nr. 8, where all variables are known. The PV technical specifications should be inputted by the user , while the temperature, wind speed at the considered location will be retrieved using the GPS position. The PV produced energy is divided by PV surface to obtain W/m2. Equation 10 is the empirical equation to retrieve the module temperature.
6) In line 226, there is an extra “the”.
Removed thanks.
7) The iPhone app requires the user to input the power of the PV system. Does this step greatly limit the application of the method? There is no one has been staring at the output power of the PV system. Maybe the author considers integrating the calculations into the PV system and sends forecast and warning information to users’ mobile phones via the Internet. This is just a personal idea which may be a reference for the author.
The reviewer is absolutely right. To be really effective, apps should not require too much user work, i.e. input parameters each time. Moreover, in some cases, the output power from the PV system might be unavailable. The best practice is to create a data log that directly injects the real time produced power into the app. This information has been added into the text.
Round 2
Reviewer 1 Report
The author has responded sufficiently to most of the comments given to the previous version. Below, please find a list of suggested corrections, which might be useful for further improving the presentation. Although the comments are provided for the newly revised portions, I believe the author can easily extend similar corrections/modifications to the rest of the paper.
L4-5 “in first instance” – “in the first instance”
L6-7 “atmospheric constituents, i.e. clouds, ...” – “atmospheric constituents, i.e., clouds, ...”
L16 “AOD value (at 550 nm) for which ...” – “AOD value (at 550 nm), for which ...”
L19 “on voluntary basis” – “on a voluntary basis”
L20 “and more importantly to monitor ... “ – “and more importantly, to monitor ...”
L26, 29, etc. “e.g.” – “e.g.,”
L28 “desert dust and biomass burning products just to name few. “ – “desert dust, and biomass burning products, just to name a few. “
L28-29 “from chemical reaction of gases” – “from the chemical reaction of gases“
L30 “at a lesser extent,” – “to a lesser extent,”
L31 “As indirect effect” – “As an indirect effect”
L32 “aerosols play a crucial crucial role” – “aerosols play a crucial role”
...
L98 “Pairwise Liu et al. [25]” – “Pairwise, Liu et al. [25]”
L100 “AOD importance in forecasting the correct panel power and its linear correlation with the available solar radiation at surface is pointed out also by Gueymard [? ]” – “The importance of AOD in forecasting the correct panel power and its linear correlation with the available solar radiation at surface were pointed out also by Gueymard et al. [38].”
L102 “in reducing the energy production” – “in reducing energy production”
L102 “results from [26], highlight that” – “results from Zhang et al. [26] highlight that”
L104 “0.15–0.31kWh m-2 per day” – “0.15–0.31 kWh m-2 per day”
...
L117 “a user defined threshold” – “a user-defined threshold”
L118 “can be considered the phase zero of more wide project” – “can be considered phase zero of a wider project”
...
L132 “I_d and I_diff are ...”: No indent should given, and “Here, I_d and I_diff are ...” is better describing the relation between this sentence and eq. (2). (It is better to avoid placing variables at the top of a sentence.)
L138 “e.g.” – “e.g.,”
L139 “1000 W/m2” – “1000 Wm-2” (the unit should be non-italic, twice) (The same applies to the caption in Fig. 3.)
L144 “G’ is an adimensional value, ...” – “Here, G’ is an non-dimensional value, ...” (no indent)
...
L153 “The theoretical computed value is compared with the real measured power P^real_out and if ...” – “The computed value is compared with the real measured power P^real_out, and if ...”
L154 “the absolute difference among the two values” – “the absolute difference between the two values”
L154-155 “the process stops and the AOD recorded.” – “the process stops and the AOD is recorded.”
L155 “The algorithm together with the FLG code is very efficient and will run under a dedicated powerful server ...” – “The algorithm, together with the FLG code, is very efficient and will run under a powerful, dedicated server ...”
L158 “In a previous study by Campbell et al. [37] more than 15000 iterations were ...” – “In a previous study by Campbell et al. [37], more than 15000 iterations were ...”
L159 “atmosphere layer” – “atmospheric layer”
...
L165 “20m a.s.l.” – “20 m a.s.l.”
L168 “in winter the average daily sunshine is ...” – “in winter, the average daily sunshine is ...”
L171 “During spring expected rain is ...” – “During spring, the expected rain is ...”
L172 “Sky is overcast ...” – “The sky is overcast ...”
...
L179 Fig. 2: the scale bar (50 m?) and direction indicator are too small. It is better to show the latitude as well.
...
L196 “To assess the accuracy of the proposed methodology before proceeding with AOD validation and retrieval, a sensitivity study ...” – “We assess the accuracy of the proposed methodology before proceeding with AOD validation and retrieval. A sensitivity study ...”
L197 “A sensitivity study of the FLG model respect to the input parameters is performed.” – “A sensitivity study is performed on the FLG model by varying the input parameters.” (The repeated use of “with respect to” should be avoided.)
L199, 201, 211, 222, 238, etc. “e.g.” – “e.g.,”
L201 “advection of different aerosol species with respect to those established during the setup” – “advection of the aerosol species different from those established during the setup” (to eliminate “with respect to”)
L205 “The AM it is exactly determined by ...” – “ The AM is exactly determined by ...”
L206-207 “The site elevation is also determined to correct compute the solar radiation. On contrary, ...” – “The site elevation is also determined to compute the solar radiation correctly. On the contrary, ...”
L208 “is assessed and results are shown in ...” – “is assessed, and results are shown in ...”
L209 “The FLG model sensitivity is assessed with respect to an ideal case ...” – “The FLG model sensitivity is assessed against an ideal case ...”
L211 “Errors due to ii) are assessed computing the AOD ...” – “Errors due to ii) are assessed by computing the AOD ...”
L212 “As shown in Figure 4 the default profile contains ...” – “As shown in Figure 4, the default profile contains
L213 “The analysis put in evidence that ii) introduces a large positive bias in the computation, above 20%.” – “The analysis reveals that the cause ii) introduces a large positive bias in the computation, above 20%.”
L214 “This large uncertainty in AOD retrieval is due to the large discrepancies between ...” – “This considerable uncertainty in AOD retrieval is due to the large discrepancies between ...” (to eliminate the first “large”)
L215-217 “In view of this result, in the new algorithm releases the atmospheric vertical profile, i.e., and then the water vapor mixing ratio atmospheric profile, will be integrated from the global meteorological models.” – Please rewrite this part. Who will release the new algorithm, and when? It is difficult to follow the logic of this part, especially “i.e., and then”. Besides, “atmospheric vertical profile” and “atmospheric profile” are duplicated. A simplified modification will be something like, “This difficulty will be alleviated by incorporating the vertical profile of the water vapor mixing ratio from the global meteorological models.”
L218-219 “assuming the presence of a cirrus cloud with an optical depth of 0.05 with 500 m thickness with the cloud base at 10 km.” – “assuming the presence of a cirrus cloud with an optical depth of 0.05, a geometrical thickness of 500 m, and the cloud base at 10 km.”
L219 “The analysis put in evidence that ...” – “The analysis indicates that ...”
L224 “In the new release a look-up-table will be implemented to input the correct albedo value using also satellite observations. “ – “In the new release, a look-up table will be implemented to input the correct albedo value, also using satellite observations.”
L225 “Photovoltaic panel technical characteristic deterioration will ...” – “The characteristics deterioration of the photovoltaic panel will ...”
L229 “In this section are presented the AOD retrievals ...” – “In this section, we present the AOD retrievals ...”
L230 “less or equal two okta . As pointed out in [39], ...” – “less or equal to two okta. As pointed out in Chew et al. [39], ...”
L231 “invisible tiny cirrus clouds might be still present” – “invisible, tiny cirrus clouds might still be present”
L232 “by those icy thin clouds. “ – “by those thin ice clouds.”
L234 “from clear-sky to moderate polluted.” – “from clear-sky to moderately polluted.”
L236 “from bigger industrial plants around.” – “from large-scale industrial plants in the vicinity.”
L238 “the local background aerosol species) during the installation and for this study is setup as urban-continental.” – “the local background aerosol species during the installation; for this study, the setup type is urban-continental.”
L240 “... has been developed for Apple Iphone models and it is shown in Figure 7.” – “... has been developed for Apple iPhone models shown in Figure 7.”
L251 “are needed.Nevertheless, “ – “are needed. Nevertheless,”
L253-255 “... conditions. To reduce the error sources listed in previous section 4.1, the vertical profile of the meteorological variables at the desired location will be obtained from global meteorological models as the European Medium-RangeWeather Forecast (ECMWF).” – “... conditions. To reduce the error sources listed in previous section 4.1, we obtain the vertical profile of the meteorological variables at the desired location from global meteorological models such as the European Medium-Range Weather Forecast (ECMWF).”
...
L257 “The presented methodology based on a reverse-engineering process, retrieves the AOD from ...” – “The presented methodology, based on a reverse-engineering process, retrieves the AOD from ...”
L260 “with a RMSE of” – “with an RMSE of”
L260-262 “The presence of thin cirrus clouds, incorrect meteorological variable atmospheric profiles and aerosol miss-typing can lead to important bias in the retrieval up to 25%. “ – “The presence of thin cirrus clouds, incorrect meteorological variables including vertical profiles of PW, and aerosol miss-typing can lead to a sizable bias in the retrieval up to 25%.”
L262-263 “the presented proof of concept part of a larger project where ...” – “the presented proof of concept is part of a larger project where ...”
L264 “it will be developed a smart-phone application to collect ...” – “a smartphone application will be developed to collect ...”
Author Response
I would like again to greatly thanks Reviewer Nr. 1. I appreciate a lot her/his work in carefully reading again the manuscript. I meticulously took into account all the suggestions in preparing the new draft.
I have just one comment. I removed the side view from Figure 2. I was not able to save the new image with marked latitude/longitude, which is now reported in the caption.